# Estimating diabetes mellitus incidence using health insurance claims data: A database-driven cohort study

**Susumu Kunisawa**[1]*, **Kyoko Matsunaga**[1], **Yuichi Imanaka**[1,2]

**1** Department of Healthcare Economics and Quality Management, Graduate School of Medicine, Kyoto University, Kyoto City, Kyoto, Japan, **2** Department of Health Security System, Centre for Health Security, Graduate School of Medicine, Kyoto University, Kyoto City, Kyoto, Japan

\* susumu.kunisawa.2v@kyoto-u.ac.jp

## Abstract

Type II diabetes mellitus is a global public health challenge, necessitating robust epidemiological investigations. The majority of evidence reports prevalence as estimations of incidence requiring longitudinal cohort studies that are challenging to conduct. However, this has been addressed by the secondary use of existing health insurance claims data. The current study aimed to examine the incidence of type II diabetes mellitus using existing claims and ledger data. The National Health Insurance and medical care system databases were used to extract type II diabetes mellitus (defined as ICD10 codes E11$–14$) claims data over a period of 5 years for individuals over 40 years old living in one city in Japan. Prevalence was calculated, and insured individuals whose data could be tracked over the entire study period were included in the subsequent analyses. Therefore, annual incidence was calculated by estimating differences in prevalence by year. Data analyses were stratified by sex and age group, and a model analysis was conducted to account for these variables. Overall, the prevalence, diabetes medication usage, and insulin usage were 26.3%, 12.1%, and 2.0%, respectively. Annual incidence of type II diabetes mellitus ranged between 1.2% and 4.6%. Both prevalence and incidence tended to be higher in males and peaked around 60–80 years old. The overall annual incidence was estimated at 3.03% (95% CI: 2.21%–3.85%). The annual incidence was not always associated with a low risk, indicating a consistent risk from middle age onward, although the level of risk varied with age. The current study successfully integrated existing claims and ledger data to explore incidence, and this methodology could be applied to a range of injuries and illnesses in the future.

## Introduction

The increasing prevalence of type II diabetes mellitus has made it a global public health challenge [1]. In Japan, existing prevention measure are primarily aimed at decreasing the risk of diabetes-related complications and death, as formulated in The Third Healthy Japan 21 [2], and development of a better understanding of disease incidence is essential to further advance these strategies.

**Data Availability Statement:** The data generated and analyzed during this study cannot be shared publicly, due to the Ethical Guidelines for Medical and Biological Research Involving Human Subjects established jointly by Japanese ministries.

Contracts were signed with the municipalities from which the data was provided, including restrictions on data users. However, other researchers may send data access requests to the staff at the Office of Research Promotion, General Affairs and Planning Division, Kyoto University (E-mail: 060kensui@mail2.adm.kyoto-u.ac.jp).

**Funding:** This study was supported by Japan Society for the Promotion of Science (grant number 20K18961 by SK and grant number 23H00448 by YI). The funders had no role in study design, data collection and analysis, decision to publish, or preparation of the manuscript.

**Competing interests:** The authors have declared that no competing interests exist.

While cross-sectional studies typically report prevalence, longitudinal cohort studies are more suitable for exploring incidence. However, they are also more challenging to conduct. A recent meta-analysis of multiple cohort studies [3] found that the incidence of type II diabetes mellitus was approximately 8.8 (95% CI: 7.4–10.4) per 1000 person-years in Japan, although large variations in the original data were observed (2.3–52.6). Other popular methodologies used include calculation of incidence using prevalence [4]. Unlike these studies, database-driven cohort studies have been attempted using existing databases. A previous Korean study successfully determined incidence using a claims database [5] Their success was largely attributed to the presence of individual IDs in the Health Insurance Review and Assessment database, which has near-total coverage in Korea.

In Japan, medical and long-term care health insurance claims data accumulate in the NDB database (National Database of Health Insurance Claims and Specific Health Checkups of Japan), and the internal structure improved by including other data. However, it is limited by the inability to identify individuals using their identification numbers, and analysis to track data on an individual basis is difficult. While cross-sectional analyses are often challenging, longitudinal analyses present even greater difficulties. In Japan, insurance databases within each insurer's unit are highly traceable over time. We believe that these databases can be used to conduct relatively robust database-driven cohort studies, although they are not nationwide like the NDB.

Therefore, the current study aimed to estimate the annual incidence rates of type II diabetes mellitus in one city in Japan using not only existing claims but also ledger databases that included all insured individuals.

## Materials and methods

This is a data-driven retrospective cohort study utilizing health insurance claims data from the National Health Insurance and medical care system of Saga City. The dataset is formally created and operated. A copy of this dataset was provided for research use under an agreement dated April 1, 2020.

Type II diabetes mellitus (defined as ICD10 codes E11\$–14\$) and insulin and diabetes medication prescription claims data for individuals over 40 years old living in the city of Saga (population: approximately 230,000) between fiscal year (FY) 2015 and FY2019 were extracted. Suspected diagnoses were not included. A health insurance claim in Japan is an invoice used by healthcare institutions to bill insurers and other responsible parties for the portion of medical costs not covered by the patient. A breakdown of medical or prescription fees being claimed is included in the invoice.

Prevalence, utilization of diabetes medication (drugs with the first three digits of the Japanese drug price reference code 396, listed in S1 File), and insulin utilization were calculated by 5-year age groups per FY based on their age in FY2015. Subsequent analyses included insured individuals who could be continuously tracked for 5 years from FY2015 to FY2019, excluding those who left the insurance system for reasons such as moving or had gaps in their tracking data, while still including those who died during this period. The continuity of individual persons was determined using claims data and ledger data for the insured persons. The ledger data records the insured persons held by the insurer and also includes insured persons who do not use healthcare. Type II diabetes mellitus claims per FY from FY2015 were then examined to determine disease status. For example, a patient who was 63 years old and had no diagnosis (i.e., type II diabetes mellitus) claim in FY2015, one diagnosis claim in FY2016, and no diagnosis claim again in FY2017 would be included in the 60–64 year age group and have a disease

status of no, yes, and yes for the respective years. This algorithm was based on the assumption that type II diabetes mellitus cannot be fundamentally cured.

Thereafter, the prevalence of type II diabetes mellitus "up to" each FY was calculated. The denominator for each year was calculated using the constant number of cases in the population in 2015 for both years, whereas the numerator for each year was the number of cases for each year. An increase in prevalence was considered the annual incidence. This straightforward method corresponds to the slope of the graph for prevalence up to each FY, estimated using linear regression (Microsoft Excel 2022, slope function). Finally, a generalized linear model weighted by the denominator was constructed to analyze the annual increase in prevalence, considering age, sex, and their interaction (R 4.4.1). Confidence intervals for the coefficient was estimated using standard errors derived from the weighted model.

The study was approved by the Kyoto University Graduate School and Faculty of Medicine, Ethics Committee (R0438), and the need for consent was waived as all data were anonymized and provided by the municipality.

## Results

The databases used in the current study included the majority of individuals >75 years of age and approximately 20% of those aged <74 years living in Saga. Table 1 shows the prevalence, diabetes medication usage, and insulin usage by age group and FY, whereas Figs 1 and 2 show the prevalence by sex. Overall, the prevalence, diabetes medication usage, and insulin usage were 26.3%, 12.1%, and 2.0%, respectively. Approximately 10% of the individuals included in this analysis did not use any healthcare services during the study period. The annual incidence of type II diabetes mellitus ranged between 1.2% and 4.6%, with the highest rates being observed in males >70 years of age. Table 2 and Figs 3 and 4 show the change in prevalence by FY (from FY2015) and age group, and the slopes of the graphs shown in these figures were used to calculate annual incidence (Table 3). The overall annual incidence, without categorization by sex or age, was estimated at 3.03%. The model, accounting for age and sex, also estimated the annual incidence at 3.03% (95% CI: 2.21%–3.85%).

## Discussion

The current study successfully estimated the incidence of type II diabetes mellitus in Japan to be approximately 2%–3% per year using not only insurance claims but also ledger databases.

The findings showed that the prevalence of type II diabetes mellitus of 26.3% was similar to the 24.2% reported in previous studies [6], despite differences in the definitions of prevalence. Although these rates were higher than the previously reported rates (i.e., 9.5–9.8/1000 person-years) in South Korea [5], direct comparisons between these two regions may not be meaningful due to significant differences in their underlying environments.

The results of this study offer several important insights. First, the line graphs, which utilized data from five FY, indicated that the transition between age groups was continuous, particularly among the younger population. This pattern suggests that approximately 2%–3% of the population is at risk of developing type II diabetes mellitus. However, these findings should be approached with caution. Inaccuracies in the algorithm used to identify type II diabetes mellitus may have occurred, and the yearly fluctuations in aggregate values, as shown in Table 1, may affect the reliability of these results.

While the change in incidence over time may be interpreted as consistent continued exposure, the current study also observed variations by age wherein incidence rates were higher in individuals aged >70 years. This agreed with previous studies that reported differences in incidence rates by age [5]. These cohort studies, including the current study, were conducted over

**Table 1. Type II diabetes mellitus prevalence, diabetes medication usage, and insulin usage by fiscal year, sex, and age group.**

| Sex | Age Category | 2015 n | PR | (case) | MU | (case) | IU | (case) | 2016 n | PR | (case) | MU | (case) | IU | (case) | 2017 n | PR | (case) | MU | (case) | IU | (case) | 2018 n | PR | (case) | MU | (case) | IU | (case) | 2019 n | PR | (case) | MU | (case) | IU | (case) |
|---|---|---|---|---|---|---|---|---|---|---|---|---|---|---|---|---|---|---|---|---|---|---|---|---|---|---|---|---|---|---|---|---|---|---|---|---|---|
| Male | 40–44 | 1692 | 5.7% | (96) | 2.9% | (49) | – | – | 1599 | 6.1% | (98) | 3.0% | (48) | 0.8% | (12) | 1537 | 6.4% | (98) | 3.0% | (46) | 0.7% | (10) | 1505 | 5.7% | (86) | 3.0% | (45) | – | – | 1427 | 5.8% | (83) | 3.1% | (44) | – | – |
| | 45–49 | 1793 | 8.8% | (157) | 4.4% | (79) | 1.3% | (23) | 1759 | 8.1% | (142) | 4.3% | (76) | 1.0% | (17) | 1658 | 9.2% | (152) | 5.1% | (85) | 0.7% | (12) | 1599 | 9.4% | (150) | 4.8% | (76) | 0.6% | (10) | 1508 | 9.2% | (139) | 5.1% | (77) | 0.7% | (11) |
| | 50–54 | 1612 | 12.9% | (208) | 7.6% | (122) | 1.7% | (28) | 1565 | 14.1% | (220) | 8.1% | (126) | 1.9% | (29) | 1559 | 13.6% | (212) | 6.8% | (106) | 1.6% | (25) | 1582 | 14.1% | (223) | 7.0% | (111) | 1.3% | (21) | 1568 | 14.3% | (224) | 7.5% | (118) | 1.3% | (20) |
| | 55–59 | 1844 | 16.0% | (295) | 8.4% | (154) | 1.7% | (31) | 1682 | 17.1% | (287) | 9.1% | (153) | 2.1% | (35) | 1599 | 17.9% | (287) | 10.1% | (161) | 2.1% | (33) | 1521 | 17.0% | (259) | 10.5% | (160) | 1.6% | (24) | 1435 | 18.9% | (271) | 10.9% | (156) | 1.7% | (24) |
| | 60–64 | 2991 | 16.5% | (493) | 9.4% | (282) | 1.8% | (55) | 2754 | 19.9% | (547) | 10.6% | (293) | 2.2% | (60) | 2572 | 22.0% | (565) | 12.0% | (309) | 2.4% | (61) | 2393 | 24.8% | (593) | 13.9% | (332) | 2.9% | (69) | 2158 | 25.3% | (546) | 13.4% | (290) | 2.9% | (63) |
| | 65–69 | 5180 | 29.5% | (1530) | 16.3% | (844) | 2.8% | (146) | 5267 | 29.5% | (1552) | 15.9% | (840) | 3.1% | (165) | 4962 | 30.9% | (1533) | 16.7% | (831) | 3.0% | (151) | 4735 | 31.0% | (1469) | 16.9% | (799) | 3.2% | (150) | 4277 | 31.3% | (1339) | 17.3% | (742) | 2.8% | (118) |
| | 70–74 | 4376 | 36.6% | (1602) | 19.3% | (844) | 3.6% | (159) | 4279 | 36.6% | (1567) | 19.5% | (835) | 3.5% | (151) | 4599 | 36.4% | (1674) | 19.2% | (883) | 3.6% | (167) | 4924 | 37.0% | (1820) | 19.3% | (950) | 3.1% | (151) | 5247 | 37.5% | (1966) | 20.1% | (1054) | 3.1% | (161) |
| | 75–79 | 5008 | 39.3% | (1970) | 21.0% | (1053) | 3.1% | (157) | 5122 | 40.2% | (2058) | 20.6% | (1056) | 3.5% | (177) | 5169 | 40.5% | (2092) | 21.4% | (1108) | 3.7% | (192) | 5364 | 42.0% | (2253) | 21.3% | (1140) | 3.5% | (186) | 5155 | 42.0% | (2163) | 21.8% | (1122) | 3.6% | (184) |
| | 80–84 | 3830 | 38.6% | (1477) | 18.6% | (711) | 3.2% | (123) | 3879 | 39.9% | (1547) | 20.0% | (776) | 3.4% | (131) | 3904 | 40.3% | (1573) | 20.0% | (780) | 3.1% | (120) | 3853 | 41.9% | (1616) | 21.1% | (813) | 3.0% | (116) | 3822 | 43.3% | (1655) | 20.8% | (795) | 3.0% | (115) |
| | 85–89 | 2282 | 34.5% | (788) | 15.6% | (356) | 2.3% | (53) | 2369 | 35.7% | (845) | 16.9% | (400) | 2.8% | (67) | 2482 | 37.9% | (941) | 16.8% | (418) | 2.8% | (69) | 2468 | 38.9% | (960) | 17.7% | (437) | 2.4% | (59) | 2519 | 40.0% | (1008) | 18.8% | (473) | 2.6% | (65) |
| | 90–94 | 887 | 31.1% | (276) | 14.1% | (125) | 1.7% | (15) | 968 | 33.6% | (325) | 13.3% | (129) | 1.5% | (15) | 1018 | 31.5% | (321) | 12.8% | (130) | 2.2% | (22) | 1052 | 32.1% | (338) | 12.9% | (136) | 2.8% | (29) | 1104 | 34.0% | (375) | 13.0% | (143) | 2.6% | (29) |
| | 95– | 184 | 22.8% | (42) | 6.5% | (12) | – | – | 203 | 29.1% | (59) | 6.9% | (14) | – | – | 238 | 29.8% | (71) | 8.4% | (20) | – | – | 279 | 30.8% | (86) | 9.3% | (26) | – | – | 285 | 29.5% | (84) | 9.8% | (28) | – | – |
| Female | 40–44 | 1585 | 2.9% | (46) | 1.5% | (24) | – | – | 1431 | 3.4% | (49) | 1.7% | (25) | – | – | 1372 | 3.0% | (41) | 1.5% | (20) | 0.7% | (10) | 1333 | 3.5% | (46) | 1.4% | (18) | – | – | 1219 | 3.0% | (36) | 1.6% | (19) | – | – |
| | 45–49 | 1592 | 5.1% | (81) | 2.1% | (34) | – | – | 1610 | 5.4% | (87) | 1.9% | (31) | – | – | 1529 | 5.7% | (87) | 2.6% | (39) | 0.7% | (11) | 1460 | 5.2% | (76) | 2.1% | (30) | – | – | 1387 | 5.2% | (72) | 2.5% | (34) | – | – |
| | 50–54 | 1665 | 6.8% | (114) | 2.8% | (47) | 0.6% | (10) | 1599 | 6.8% | (109) | 2.6% | (41) | 0.6% | (10) | 1560 | 6.9% | (107) | 2.6% | (40) | 0.6% | (10) | 1534 | 7.0% | (108) | 3.3% | (51) | 0.7% | (12) | 1424 | 8.4% | (120) | 3.7% | (52) | – | – |
| | 55–59 | 2060 | 10.9% | (224) | 4.1% | (85) | 1.0% | (21) | 1871 | 11.4% | (213) | 4.1% | (76) | 0.9% | (16) | 1766 | 10.6% | (188) | 4.6% | (82) | 0.9% | (16) | 1668 | 10.9% | (181) | 4.4% | (73) | 0.8% | (13) | 1635 | 11.3% | (185) | 4.6% | (76) | 0.8% | (13) |
| | 60–64 | 3729 | 12.8% | (476) | 5.4% | (200) | 0.8% | (29) | 3507 | 13.2% | (462) | 5.6% | (197) | 0.9% | (31) | 3163 | 14.2% | (449) | 5.7% | (179) | 1.2% | (37) | 2966 | 15.5% | (460) | 6.4% | (191) | 1.2% | (37) | 2742 | 16.3% | (447) | 6.7% | (185) | 1.7% | (46) |
| | 65–69 | 6336 | 21.8% | (1382) | 9.4% | (595) | 1.4% | (86) | 6287 | 21.9% | (1379) | 9.2% | (576) | 1.4% | (88) | 6075 | 22.0% | (1337) | 9.2% | (561) | 1.4% | (84) | 5725 | 21.8% | (1249) | 8.9% | (507) | 1.2% | (70) | 5270 | 21.3% | (1123) | 8.9% | (469) | 1.3% | (71) |
| | 70–74 | 5685 | 27.5% | (1561) | 11.7% | (665) | 1.5% | (88) | 5588 | 28.9% | (1613) | 12.2% | (681) | 1.6% | (89) | 5774 | 28.6% | (1654) | 12.1% | (698) | 1.4% | (79) | 6070 | 28.6% | (1737) | 11.8% | (718) | 1.4% | (86) | 6357 | 29.2% | (1854) | 11.9% | (756) | 1.6% | (99) |
| | 75–79 | 6690 | 28.9% | (1932) | 12.8% | (853) | 2.0% | (137) | 6969 | 30.4% | (2117) | 13.1% | (910) | 2.0% | (137) | 6984 | 31.9% | (2229) | 13.1% | (918) | 2.0% | (140) | 7230 | 32.2% | (2331) | 13.3% | (960) | 1.8% | (129) | 6997 | 32.2% | (2256) | 13.7% | (959) | 1.6% | (110) |
| | 80–84 | 6042 | 27.5% | (1660) | 11.8% | (715) | 2.0% | (121) | 5952 | 28.9% | (1723) | 12.2% | (728) | 2.2% | (129) | 5845 | 30.1% | (1759) | 12.4% | (725) | 1.8% | (106) | 5710 | 31.5% | (1796) | 13.3% | (761) | 2.0% | (117) | 5592 | 33.6% | (1878) | 13.6% | (758) | 2.0% | (110) |
| | 85–89 | 4420 | 25.4% | (1121) | 10.8% | (477) | 1.9% | (84) | 4564 | 25.6% | (1168) | 11.0% | (500) | 1.8% | (82) | 4811 | 26.9% | (1295) | 10.8% | (518) | 1.6% | (78) | 4767 | 27.8% | (1327) | 10.6% | (503) | 1.4% | (66) | 4810 | 29.2% | (1405) | 11.4% | (547) | 1.5% | (72) |
| | 90–94 | 2358 | 21.3% | (503) | 7.7% | (182) | 1.3% | (30) | 2453 | 21.7% | (532) | 8.4% | (207) | 1.1% | (28) | 2548 | 24.3% | (620) | 9.3% | (238) | 1.2% | (31) | 2669 | 25.4% | (679) | 10.0% | (267) | 1.7% | (46) | 2802 | 26.2% | (735) | 9.2% | (259) | 1.4% | (39) |
| | 95– | 914 | 13.8% | (126) | 3.3% | (30) | – | – | 982 | 14.1% | (138) | 4.1% | (40) | – | – | 1016 | 16.2% | (165) | 4.4% | (45) | 1.1% | (11) | 1079 | 17.4% | (188) | 4.4% | (47) | – | – | 1147 | 18.1% | (208) | 5.7% | (65) | 1.0% | (11) |

Cells with a small number of cases (i.e., <10) have been masked (using -) to ensure anonymity.

n: number of subjects

PR: prevalence

MU: medication utilization

IU: insulin utilization

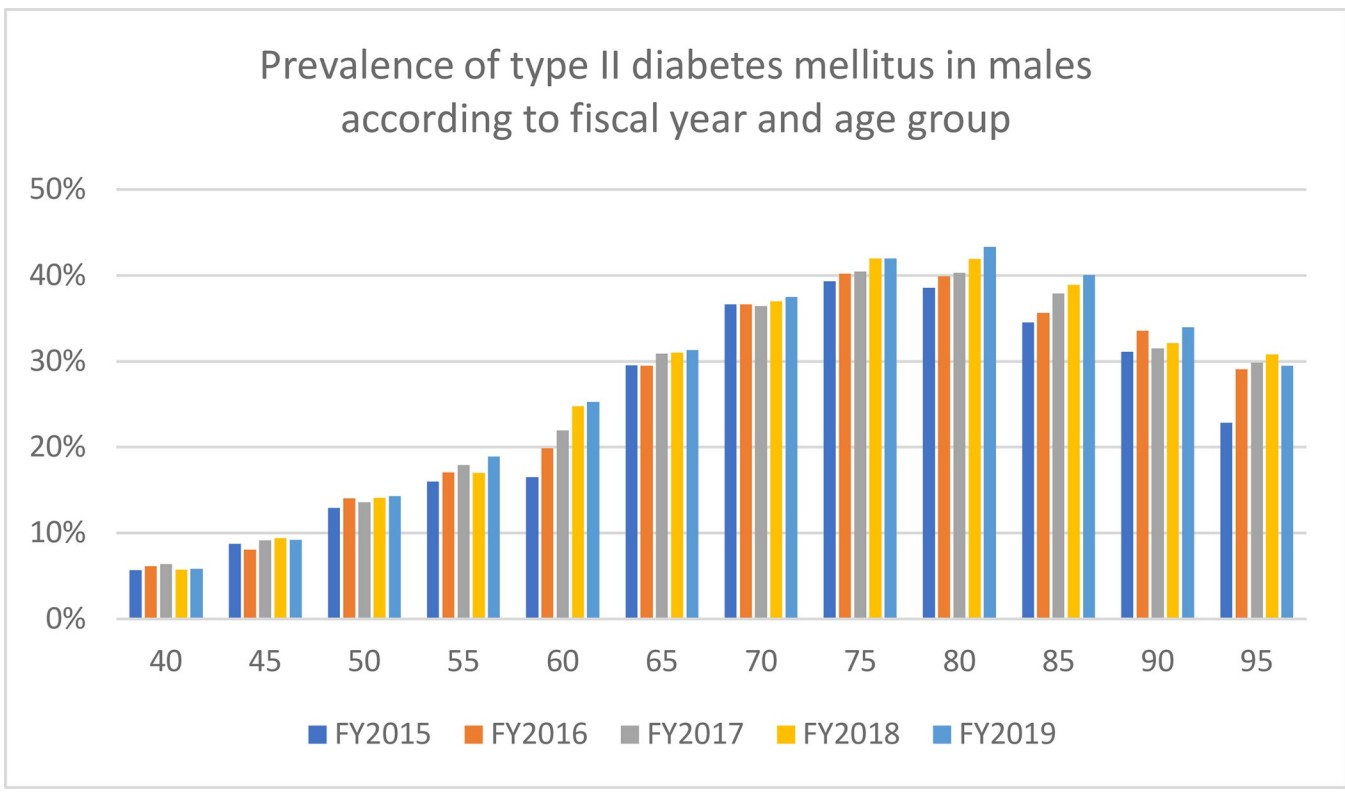

**Fig 1. Prevalence of type II diabetes mellitus in males according to fiscal year and age group.**

a relatively short period (5 years), making it difficult to consider the impact of death as a long-term transition over life. Although a true cohort would include those who died in the course of their lives up to the time of their death, a limited set of data obtained from 5 years cannot include those who have already died before that time period. Therefore, caution must be exercised in interpreting the exclusion of cases that have already died by a time period that is not included in this database.

The database-driven methodology proposed here can potentially be used to explore the incidence rates of other conditions including diabetes-related complications such as myocardial infarction and stroke; injuries; and other diseases such as hyperlipidemia or cancer. Furthermore, it can also be used to examine precise populations including patients diagnosed with type II diabetes mellitus and treated using specific drugs such as those examined in the current study.

The effects of various factors can also be analyzed by arbitrarily dividing the background populations into separate groups for tracking and analysis and complementing available data by combining additional databases. General cohort studies are limited by the inability to consider additional factors after commencement, and this can be addressed to a certain extent by cohort studies that combine datasets.

The methodologies used in a previous study [5] conducted in Korea are based on the assumption that the majority of the population are covered by one claims database. This suggests that findings from the small number of countries where this is not applicable maybe biased. However, in the current study, the cohort was created and tracked using ledger data, which made observation and calculation considerably simpler.

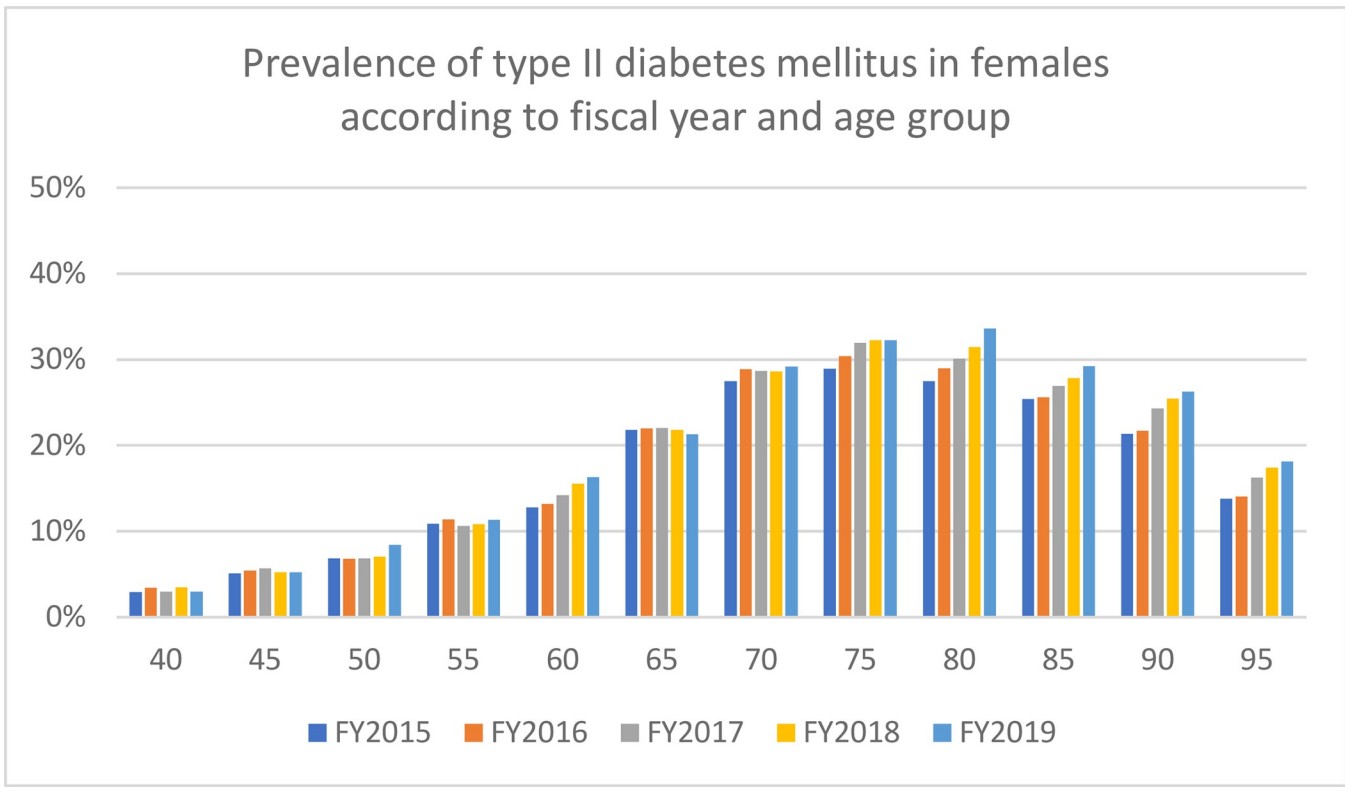

**Fig 2. Prevalence of type II diabetes mellitus in females according to fiscal year and age group.**

One limitation of this study is that the data are not all-inclusive for the entire population. A proportion of residents under 75 years of age are included. The main reason for the loss of follow-up during the study period was changes in insurance, including moving. Of note, our analysis included cases of mortality occurring during the study period, suggesting that the actual prevalence and incidence rates may be higher. While this study primarily focused on relatively straightforward analyses, more complex approaches, such as survival analysis, may also be considered, depending on the study's context and objectives. Although we believe that the results will not be significantly distorted by these limitations, we cannot ensure that the figures are definitive based on this study alone. However, the methodology is useful.

Another limitation of the current methodology was that the case identification algorithm used relied heavily on the claims diagnoses, preventing identification of overdiagnosis or misdiagnosis events; patients with diabetes that went unnoticed; and latent diabetes cases that did not undergo treatment. However, this limitation was related to the definition of diabetes used, and other applications of this method such as examination of the incidence of diabetes medication usage would likely provide more robust results.

In Japan, it is impossible to track individuals across different databases using their personal ID numbers, particularly if they change their medical insurance. Therefore, the resultant cohorts are relatively small despite the creative approach used in this study. However, as the identification system in Japan (i.e., My Number) expands and the claims database is adapted accordingly, the actual incidence of the disease will become more apparent. Meanwhile, incorporation of ledger and claims data is recommended for studies aiming to examine incidence rates using existing datasets.

**Table 2. Type II diabetes mellitus prevalence by fiscal year (from 2015 to each subsequent year) by sex and age group.**

| | Fiscal Year | | 2015 | | From 2015 to 2016 | | From 2015 to 2017 | | From 2015 to 2018 | | From 2015 to 2019 | |
|---|---|---|---|---|---|---|---|---|---|---|---|---|
| | Age Category at FY2015 | n | PR | (case) | PR | (case) | PR | (case) | PR | (case) | PR | (case) |
| Male | 40–44 | 1148 | 5.8% | (67) | 7.5% | (86) | 9.4% | (108) | 11.2% | (129) | 13.0% | (149) |
| | 45–49 | 1303 | 9.2% | (120) | 11.5% | (150) | 13.7% | (178) | 15.5% | (202) | 18.1% | (236) |
| | 50–54 | 1188 | 13.5% | (160) | 17.4% | (207) | 20.8% | (247) | 23.4% | (278) | 25.8% | (307) |
| | 55–59 | 1468 | 16.5% | (242) | 19.9% | (292) | 22.9% | (336) | 25.3% | (372) | 28.1% | (412) |
| | 60–64 | 2559 | 17.4% | (445) | 23.2% | (594) | 28.1% | (719) | 32.2% | (823) | 36.0% | (921) |
| | 65–69 | 4781 | 29.8% | (1424) | 34.1% | (1629) | 38.3% | (1830) | 41.5% | (1986) | 44.7% | (2135) |
| | 70–74 | 2754 | 37.1% | (1023) | 41.2% | (1136) | 45.0% | (1239) | 48.3% | (1330) | 50.5% | (1391) |
| | 75–79 | 4528 | 38.7% | (1754) | 43.3% | (1959) | 47.2% | (2136) | 50.4% | (2282) | 53.0% | (2402) |
| | 80–84 | 3684 | 38.8% | (1428) | 42.8% | (1576) | 45.6% | (1681) | 48.7% | (1794) | 51.0% | (1877) |
| | 85–89 | 2180 | 34.5% | (752) | 38.5% | (840) | 41.4% | (903) | 43.7% | (952) | 45.9% | (1000) |
| | 90–94 | 824 | 30.8% | (254) | 35.6% | (293) | 37.5% | (309) | 39.3% | (324) | 40.0% | (330) |
| | 95– | 174 | 23.6% | (41) | 29.3% | (51) | 29.3% | (51) | 30.5% | (53) | 31.6% | (55) |
| Female | 40–44 | 911 | 3.2% | (29) | 4.6% | (42) | 5.9% | (54) | 7.2% | (66) | 7.9% | (72) |
| | 45–49 | 1008 | 6.0% | (60) | 8.5% | (86) | 10.1% | (102) | 12.0% | (121) | 13.4% | (135) |
| | 50–54 | 1175 | 7.3% | (86) | 9.8% | (115) | 11.1% | (130) | 12.5% | (147) | 14.0% | (164) |
| | 55–59 | 1604 | 11.5% | (185) | 14.8% | (237) | 17.1% | (275) | 19.1% | (307) | 21.5% | (345) |
| | 60–64 | 3250 | 12.5% | (406) | 16.5% | (537) | 20.2% | (656) | 23.0% | (747) | 26.0% | (846) |
| | 65–69 | 5829 | 22.3% | (1297) | 26.4% | (1539) | 29.9% | (1741) | 32.7% | (1906) | 35.1% | (2044) |
| | 70–74 | 3344 | 28.3% | (948) | 31.9% | (1068) | 35.4% | (1183) | 38.1% | (1275) | 40.8% | (1366) |
| | 75–79 | 5977 | 29.0% | (1735) | 33.0% | (1970) | 36.8% | (2197) | 39.8% | (2378) | 42.6% | (2549) |
| | 80–84 | 5738 | 27.5% | (1577) | 31.3% | (1796) | 34.3% | (1970) | 36.8% | (2112) | 39.4% | (2259) |
| | 85–89 | 4142 | 25.5% | (1058) | 28.2% | (1170) | 31.2% | (1292) | 33.6% | (1393) | 36.1% | (1494) |
| | 90–94 | 2189 | 21.2% | (463) | 23.9% | (523) | 26.8% | (587) | 28.1% | (615) | 29.6% | (648) |
| | 95– | 859 | 13.7% | (118) | 15.7% | (135) | 17.7% | (152) | 18.5% | (159) | 19.3% | (166) |

n: number of subjects

FY: fiscal year

PR: Prevalence by FY from FY2015.

## Conclusion

This database-driven cohort study of individuals aged >40 years examined the incidence of type II diabetes mellitus in Japan. The incidence rate was approximately 2%–3% per year, with the risk of developing the disease being continuous from middle age onward.

**Table 3. Annual incidence rates of type II diabetes mellitus.**

| Age Category at Fiscal Year 2015 | 40–44 | 45–49 | 50–54 | 55–59 | 60–64 | 65–69 | 70–74 | 75–79 | 80–84 | 85–89 | 90–94 | 95– |
|---|---|---|---|---|---|---|---|---|---|---|---|---|
| Male | 1.80% | 2.18% | 3.07% | 2.86% | 4.62% | 3.72% | 3.38% | 3.58% | 3.03% | 2.79% | 2.22% | 1.72% |
| Female | 1.21% | 1.84% | 1.60% | 2.43% | 3.35% | 3.19% | 3.12% | 3.41% | 2.93% | 2.64% | 2.11% | 1.40% |

Annual incidence is calculated as the slopes of the increases in prevalence

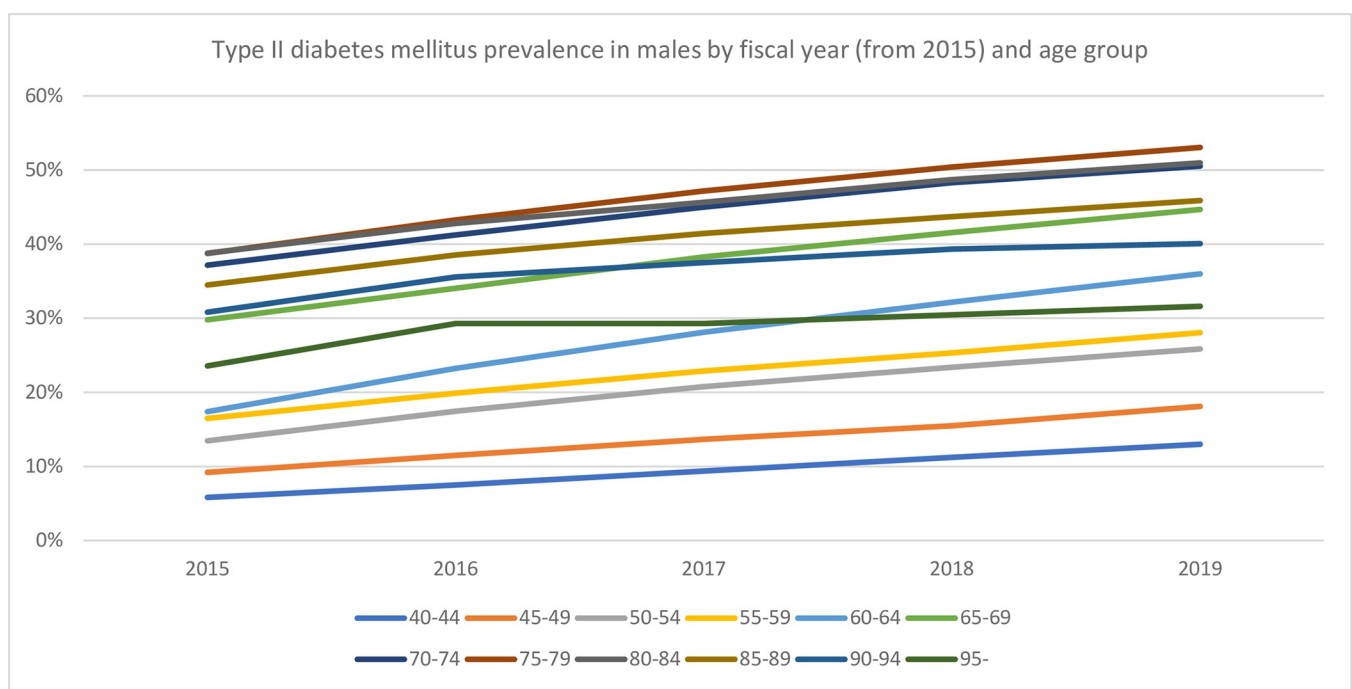

**Fig 3. Type II diabetes mellitus prevalence in males by fiscal year (from 2015) and age group.**

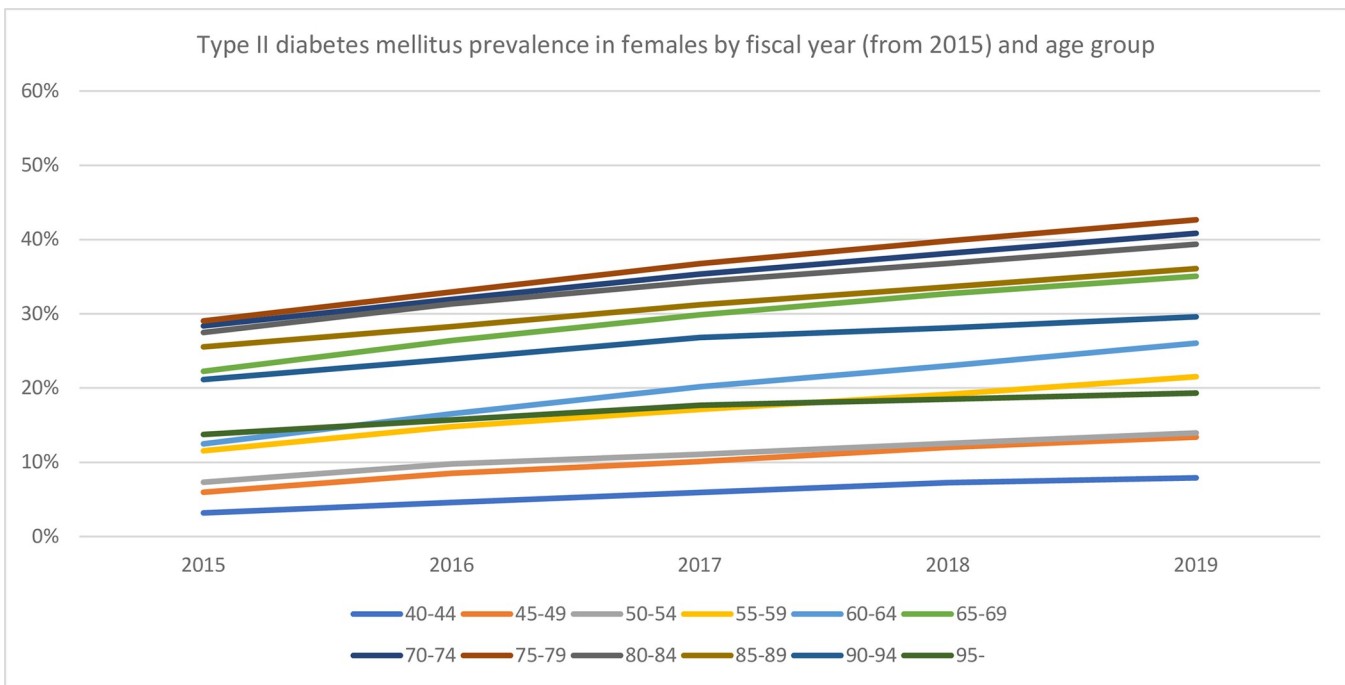

**Fig 4. Type II diabetes mellitus prevalence in females by fiscal year (from 2015) and age group.**

## Supporting information

**S1 File. List of diabetes medications identified in this study.** Drugs with the first three digits of the Japanese drug price reference code 396.
(CSV)

## Acknowledgments

We sincerely thank the individuals at Saga City Office for their collaboration and assistance with this research.

## Author Contributions

**Conceptualization:** Susumu Kunisawa.

**Formal analysis:** Susumu Kunisawa.

**Funding acquisition:** Susumu Kunisawa.

**Investigation:** Susumu Kunisawa.

**Methodology:** Susumu Kunisawa.

**Project administration:** Yuichi Imanaka.

**Supervision:** Yuichi Imanaka.

**Writing – original draft:** Susumu Kunisawa.

**Writing – review & editing:** Kyoko Matsunaga.

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
