## [Decision Letter · Decision Letter 0]

29 Jul 2024

PONE-D-24-14955Estimating diabetes mellitus incidence using medical reimbursement data: A database-driven cohort studyPLOS ONE

Dear Dr. Kunisawa,

Thank you for submitting your manuscript to PLOS ONE. After careful consideration, we feel that it has merit but does not fully meet PLOS ONE’s publication criteria as it currently stands. Therefore, we invite you to submit a revised version of the manuscript that addresses the points raised during the review process.

 Please  re-analyze the data in order to included age group, sex, and year by using modeling

We look forward to receiving your revised manuscript.

Kind regards,

Hamid Reza Baradaran, M.D., Ph.D.,

Academic Editor

PLOS ONE

2. For studies involving third-party data, we encourage authors to share any data specific to their analyses that they can legally distribute. PLOS recognizes, however, that authors may be using third-party data they do not have the rights to share. When third-party data cannot be publicly shared, authors must provide all information necessary for interested researchers to apply to gain access to the data. (https://journals.plos.org/plosone/s/data-availability#loc-acceptable-data-access-restrictions)

a) A description of the data set and the third-party source

b) If applicable, verification of permission to use the data set

c) Confirmation of whether the authors received any special privileges in accessing the data that other researchers would not have

d) All necessary contact information others would need to apply to gain access to the data

Additional Editor Comments:

Please re-analyze the data in order to included age group, sex, and year by using modeling

Reviewers' comments:

Reviewer's Responses to Questions

**Comments to the Author**

1. Is the manuscript technically sound, and do the data support the conclusions?

Reviewer #1: Yes

Reviewer #2: Yes

Reviewer #3: Partly

Reviewer #4: Partly

2. Has the statistical analysis been performed appropriately and rigorously? 

Reviewer #1: Yes

Reviewer #2: Yes

Reviewer #3: No

Reviewer #4: No

3. Have the authors made all data underlying the findings in their manuscript fully available?

Reviewer #1: Yes

Reviewer #2: No

Reviewer #3: Yes

Reviewer #4: Yes

4. Is the manuscript presented in an intelligible fashion and written in standard English?

Reviewer #1: Yes

Reviewer #2: Yes

Reviewer #3: Yes

Reviewer #4: Yes

5. Review Comments to the Author

Reviewer #1: This paper is to analyze Type II diabetes mellitus incidence using medical reimbursement data, as it is clearly stated in the title. Although the authors accesses limited data of a particular city, saga, out of nation-wide claim database, the statistic data tells the demographic structure of Japan. If the authors can provide some additional discussion about representativeness of the data of the city to support understandings of readers to know about the meaning of the data under comparison with some other references such as NDB open Data. Additional information about legal limitation of data sharing about NDB data should be denoted into the Acknowledgement part.

Reviewer #2: Dear Prof. Dr. Baradaran,

In this article, the authors estimated the prevalence and incidence rate of type II diabetes mellitus, diabetes medication, and insulin utilization using the National Health Insurance and Medical Care System databases. I was very pleased to see this interesting work since the health data from large and comprehensive databases are growing both in size and quality and can be used for better, cheaper, and more accurate estimation of the burden of diseases.

Funding details, data availability statement, and competing interests were disclosed correctly, authors prepared the manuscript according to the journal’s guidelines, and items of the STROBE checklist were included in the reported sections. The tables and figures are clear. However, there are some issues to be addressed and the manuscript needs major revision:

Major Issues:

1. Please include the approval number/ethics code indicating approval of this research in the ethics statement.

Minor Issues:

Introduction:

1. Line 46, page 3: The reference (2) cites the 2nd term of Health Japan 21, while in the text authors mentioned the 3rd term. Also, the URL link provided in the reference list directs to a page with a “404 Not Found” error. Please consider correcting the citation.

2. Throughout the manuscript the phrase “prevalence rate” is used, which is incorrect. Prevalence is a proportion, not a rate (because rates have a specified period of time in their denominator which prevalence lacks) and unfortunately, it is a common mistake the authors make in the scientific literature. So please correct all of the “prevalence rate” phrases to simply “prevalence” in the abstract, main body, tables, and figures. Also, this applies to diabetes medication usage and insulin usage rates.

3. Line 48, page 3: It is a fact that cross-sectional studies report prevalence and it is not necessary to cite an article for this regard. Also, the cited reference is not supporting the statement. Please consider removing the citation or adding more related citations.

4. Line 52, page 3: The cited reference reported the incidence of type II diabetes in Japan as 8.8 (95% CI: 7.4–10.4) per 1000 person-years. Please check the reference and correct it accordingly.

5. Line 53, page 3: Please update the URL link provided in reference (5) as it redirects to a page with the error “The page you're looking for was not found”.

6. Line 58, page 3: Please spell out the full term at its first mention, indicate its abbreviation in parenthesis, and use the abbreviation from then on.

Methods:

7. Please clearly mention the study design in the Methods section.

8. The words “ledger”, “reimbursement”, and “claim” can have various meanings in different countries based on their legal, fiscal, and medical systems. It would be great if you provide the exact meaning of these words according to the Japanese systems and laws in the Methods section.

9. In the Methods section, Please clearly explain how you calculated the population at risk for each fiscal year.

10. Considering that claims and ledger data have exact dates of the disease’s diagnosis, then it would be a great idea if you report the incidence rate by person-years.

11. Please indicate which medications you considered as diabetes medications.

Results:

12. Please report the annual incidence data without grouping for age and sex (i.e. in Total for all ages and all sexes). Also, please consider statistically comparing the age groups and sexes with each other for significant differences.

13. In the Results section please mention the exact numbers/percentages if you did not mention them in full in the tables or figures.

14. Please mention the number of the included and excluded individuals in the study. You can consider drawing a flowchart.

15. Line 105, page 6: In addition to fiscal year and age group, data are also grouped by sex, please add it to title of the Table 1.

16. Line 112, page 7: Like Table 1, please correct Table 2’s title and add age group and sex.

17. Line 115, page 7: You mentioned PR as the “change” in prevalence. Considering the fiscal year (FY) 2015 as the reference year, then there should not be a column dedicated to FY 2015. I highly recommend replacing this table with a similar table that shows the annual incidence rate by age group, FY, and sex.

18. Line 117, page 8: If Table 3 shows the annual incidence rate, then it should present the data for each year, if it shows the mean annual rate, then you should mention this both in the title and text. Also, Table 3’s lines are faded and not visible.

19. According to the figure titles

Discussion:

20. In the first paragraph of the Discussion section, please only summarize the main findings.

21. I do not think it is a good idea to compare the findings of a city in Japan to the national incidence rate of South Korea considering significant differences between them. It will be more suitable if you compare your findings to the articles reporting rates for other municipalities and prefectures in Japan, or even at the national level so you can assess the comparability of your findings with territories sharing more similar context.

22. The second paragraph of the Discussion section (lines 139-145) is very unclear and hard to understand. Please consider rewriting it.

Also, I am available to review the revised version as soon as the authors provide it.

Sincerely

Reviewer #3: The abstract lacks sufficient detail about the methods. In the main text, there is inadequate information on data validity, the primary data source, and the data collection process. While they included data from a proportion of participants under 75 years, the selection process and potential selection bias were not addressed. My primary concern lies in the analysis: why was a Poisson regression model not utilized to assess the effects of age group, sex, and year? They report incidence and prevalence by these variables without indicating any interactions among them. I strongly recommend modeling the data.

Reviewer #4: This study can be an important one in terms of using an insurance data base for estimating health measures. however, I have a number of comments:

Title: OK

Abstract: I think it is necessary to show the confidence intervals of the estimated measures as a proxy for precision of the estimates.

Keywords: OK; may be “big data” is not appropriate keyword for this study.

Introduction: Please show what NDB stands for. It is also necessary to express the novelty or applicability of the study in this section explicitly.

Material and method: Please notice to the following issues in this section:

- Please define the nominator and denominator used for estimating the proposed measures in detail in this section.

- It is necessary to address loss to follow up for estimating the incidence rate.

- The method of data analysis has been missed. Another issue to be noticed is the effect of new people entering to the proposed age groups and the mortality rate.

- As the last point, I think the validity of the method shall be addressed in this section as well.

Results: It is necessary to show the confidence interval as a measure of the precision of the estimates.

Discussion: OK.

References: OK

6. PLOS authors have the option to publish the peer review history of their article (what does this mean?). If published, this will include your full peer review and any attached files.

Reviewer #1: **Yes: **Tomohiro Kuroda

Reviewer #2: No

Reviewer #3: **Yes: **AliAKbar Haghdoost

Reviewer #4: **Yes: **Babak Eshrati

---

## [Author Response · Author response to Decision Letter 0]

7 Aug 2024

Response to Reviewers

Dear Hamid Reza Baradaran and Reviewers, 

Thank you for your suggestions on improving our manuscript.

As to restrictions on data sharing, the submission form includes the full information required as the following. 

a) A description of the data set and the third-party source

b) If applicable, verification of permission to use the data set

c) Confirmation of whether the authors received any special privileges in accessing the data that other researchers would not have

d) All necessary contact information others would need to apply to gain access to the data

“The data generated and analyzed during this study cannot be shared publicly, due to the Ethical Guidelines for Medical and Biological Research Involving Human Subjects established jointly by Japanese ministries. Contracts were signed with the municipalities from which the data was provided, including restrictions on data users. However, other researchers may send data access requests to Office of Research Promotion, General Affairs and Planning Division, Kyoto University (E-mail: 060kensui@mail2.adm.kyoto-u.ac.jp).”

Reviewer #1

1-1. If the authors can provide some additional discussion about representativeness of the data of the city to support understandings of readers to know about the meaning of the data under comparison with some other references such as NDB open Data.

Making comparative references between this study and the NDB Open Data, which are currently provided as relatively simple aggregate data, was difficult. Instead, we have added a reference for the differences between the handling of cross-sectional data, such as NDB Open Data, and the longitudinal data.

“While cross-sectional analyses are often challenging, longitudinal analyses present even greater difficulties.”

1-2. Additional information about legal limitation of data sharing about NDB data should be denoted into the Acknowledgement part.

The NDB, which we have cited in the document, has strict legal restrictions on using and sharing data, but this study did not use NDB. Therefore, we did not add any further references beyond the current text.

Reviewer #2

2-1. Please include the approval number/ethics code indicating approval of this research in the ethics statement.

We added the approval number.

2-2-1. Line 46, page 3: The reference (2) cites the 2nd term of Health Japan 21, while in the text authors mentioned the 3rd term. Also, the URL link provided in the reference list directs to a page with a “404 Not Found” error. Please consider correcting the citation.

This was the authors’ English translation mistake. “The third term”’ is correct. 

The URL is correctly listed in our manuscript, but the PDF generator of the submission system caused an error. 

2-2-2. Throughout the manuscript the phrase “prevalence rate” is used, which is incorrect. Prevalence is a proportion, not a rate (because rates have a specified period of time in their denominator which prevalence lacks) and unfortunately, it is a common mistake the authors make in the scientific literature. So please correct all of the “prevalence rate” phrases to simply “prevalence” in the abstract, main body, tables, and figures. Also, this applies to diabetes medication usage and insulin usage rates.

Thank you for your guidance. We changed “prevalence rate” to “prevalence” throughout the manuscript.

2-2-3. Line 48, page 3: It is a fact that cross-sectional studies report prevalence and it is not necessary to cite an article for this regard. Also, the cited reference is not supporting the statement. Please consider removing the citation or adding more related citations.

We removed this citation.

2-2-4. Line 52, page 3: The cited reference reported the incidence of type II diabetes in Japan as 8.8 (95% CI: 7.4–10.4) per 1000 person-years. Please check the reference and correct it accordingly.

Based on the cited literature, in addition to the CIs, we showed that their estimates are heterogeneous. We revised the manuscript to clarify this issue.

2-2-5. Line 53, page 3: Please update the URL link provided in reference (5) as it redirects to a page with the error “The page you're looking for was not found”.

We updated this URL.

2-2-6. Line 58, page 3: Please spell out the full term at its first mention, indicate its abbreviation in parenthesis, and use the abbreviation from then on.

We made this revision. 

Methods:

2-2-7. Please clearly mention the study design in the Methods section.

We made this revision.

2-2-8. The words “ledger”, “reimbursement”, and “claim” can have various meanings in different countries based on their legal, fiscal, and medical systems. It would be great if you provide the exact meaning of these words according to the Japanese systems and laws in the Methods section.

To clarify the notations concerning claims, we unified the terms to claim and added a description of claim and ledger.

2-2-9. In the Methods section, Please clearly explain how you calculated the population at risk for each fiscal year.

We made this revision.

2-2-10. Please indicate which medications you considered as diabetes medications.

We made this revision.

2-2-11. Results:Please report the annual incidence data without grouping for age and sex (i.e. in Total for all ages and all sexes). Also, please consider statistically comparing the age groups and sexes with each other for significant differences.

The annual incidence for all of the categories was added. Although differences between categories could be tested, this would require multiple tests, so we decided not to report the differences between categories. However, an additional statistical analysis was performed and reported to account for these groups, as suggested by the editor and other reviewers.

2-2-12. In the Results section please mention the exact numbers/percentages if you did not mention them in full in the tables or figures.

The exact numbers are marked on the table.

2-2-13. Please mention the number of the included and excluded individuals in the study. You can consider drawing a flowchart.

In this study, simple inclusion was made from the database, as indicated in the Methods, and no exclusions were made. 

2-2-14. Line 105, page 6: In addition to fiscal year and age group, data are also grouped by sex, please add it to title of the Table 1. Line 112, page 7: Like Table 1, please correct Table 2’s title and add age group and sex.

We made this revision.

2-2-15. Line 115, page 7: You mentioned PR as the “change” in prevalence. Considering the fiscal year (FY) 2015 as the reference year, then there should not be a column dedicated to FY 2015. I highly recommend replacing this table with a similar table that shows the annual incidence rate by age group, FY, and sex.

This was a mistake. “Prevalence by FY from FY2015” is correct and the present table is appropriate.

2-2-16. Line 117, page 8: If Table 3 shows the annual incidence rate, then it should present the data for each year, if it shows the mean annual rate, then you should mention this both in the title and text. Also, Table 3’s lines are faded and not visible.

As shown in the Methods, this is the calculation of slopes of the increased prevalence. We added an explanation to the Table.

2-2-17. According to the figure titles

We made this revision.

Discussion:

2-2-18. In the first paragraph of the Discussion section, please only summarize the main findings.

We made this revision.

2-2-19. I do not think it is a good idea to compare the findings of a city in Japan to the national incidence rate of South Korea considering significant differences between them. It will be more suitable if you compare your findings to the articles reporting rates for other municipalities and prefectures in Japan, or even at the national level so you can assess the comparability of your findings with territories sharing more similar context.

I agree. We retained the quotation marks but rewrote the text to clarify that the comparison of figures was not meaningful.

2-2-20. The second paragraph of the Discussion section (lines 139-145) is very unclear and hard to understand. Please consider rewriting it.

We revised the text.

Thank you very much for your detailed feedback that allowed us to improve our manuscript.

Reviewer #3: 

3-1.The abstract lacks sufficient detail about the methods. 

A model analysis was added and the methods were revised.

3-2.In the main text, there is inadequate information on data validity, the primary data source, and the data collection process. 

We revised the methods explaining the data source.

3-3.While they included data from a proportion of participants under 75 years, the selection process and potential selection bias were not addressed. 

The data did not represent a partial selection of people aged 74 years and under but represented the limited number of citizens who have this insurance. We believe that this did not affect the results, but we have added the description of this point as a limitation.

3-3. My primary concern lies in the analysis: why was a Poisson regression model not utilized to assess the effects of age group, sex, and year? They report incidence and prevalence by these variables without indicating any interactions among them. I strongly recommend modeling the data.

We consider the results very useful only in the initial results. We added the analysis using a statistical model.

Reviewer #4: 

4-1. Abstract: I think it is necessary to show the confidence intervals of the estimated measures as a proxy for precision of the estimates.

A model analysis and confidence intervals have been added to the abstract.

4-2. Keywords: OK; may be “big data” is not appropriate keyword for this study.

We removed “big data” from the Keywords.

4-3. Introduction: Please show what NDB stands for. It is also necessary to express the novelty or applicability of the study in this section explicitly.

We added an explanation of the abbreviation “NDB,” and the text was revised to clarify that this study is challenging because tracking the data at the individual level was difficult in contrast to national NDB.

4-4. Material and method: Please notice to the following issues in this section:

- Please define the nominator and denominator used for estimating the proposed measures in detail in this section.

We revised the explanation by mentioning the denominator and numerator.

4-5. It is necessary to address loss to follow up for estimating the incidence rate.

We added this point as a limitation to the study.

4-6. The method of data analysis has been missed. 

We added a model analysis and revised the methods.

4-7. Another issue to be noticed is the effect of new people entering to the proposed age groups and the mortality rate.

This study covers individuals who can be tracked for 5 years. There were no transfers from the middle of the study period. The text was revised to make this explicit. Conversely, deaths during the period were included in the population analysis, which could introduce bias. This was added as a limitation of the study.

4-8. As the last point, I think the validity of the method shall be addressed in this section as well.

The main considerations were included in the discussion, but we have also mentioned this point in the methods section.

---

## [Decision Letter · Decision Letter 1]

9 Sep 2024

PONE-D-24-14955R1Estimating diabetes mellitus incidence using health insurance claims data: A database-driven cohort studyPLOS ONE

Dear Dr. Kunisawa,

Thank you for submitting your manuscript to PLOS ONE. After careful consideration, we feel that it has merit but does not fully meet PLOS ONE’s publication criteria as it currently stands. Therefore, we invite you to submit a revised version of the manuscript that addresses the points raised during the review process.

  please   define the method of denominator estimation in more detail.

We look forward to receiving your revised manuscript.

Kind regards,

Hamid Reza Baradaran, M.D., Ph.D.,

Academic Editor

PLOS ONE

Journal Requirements:

Additional Editor Comments:

As one the reviewers has mentioned please define the method of denominator estimation in more detail.

Reviewers' comments:

Reviewer's Responses to Questions

**Comments to the Author**

1. If the authors have adequately addressed your comments raised in a previous round of review and you feel that this manuscript is now acceptable for publication, you may indicate that here to bypass the “Comments to the Author” section, enter your conflict of interest statement in the “Confidential to Editor” section, and submit your "Accept" recommendation.

Reviewer #4: All comments have been addressed

2. Is the manuscript technically sound, and do the data support the conclusions?

Reviewer #4: Yes

3. Has the statistical analysis been performed appropriately and rigorously? 

Reviewer #4: Yes

4. Have the authors made all data underlying the findings in their manuscript fully available?

Reviewer #4: Yes

5. Is the manuscript presented in an intelligible fashion and written in standard English?

Reviewer #4: Yes

6. Review Comments to the Author

Reviewer #4: I think most of the comments are addressed by the distinguished authors. However, there are a number of issues to be noticed:

- I think it is necessary to define the method of denominator estimation in more detail. This is especially true considering emigration or death which ordinarily happens in populations.

- Please define the method of confidence interval estimation.

7. PLOS authors have the option to publish the peer review history of their article (what does this mean?). If published, this will include your full peer review and any attached files.

Reviewer #4: **Yes: **Babak Eshrati

---

## [Author Response · Author response to Decision Letter 1]

13 Sep 2024

Response to Reviewers

Dear Hamid Reza Baradaran and Babak Eshrat, 

Thank you for your suggestions on improving our manuscript.

--please define the method of denominator estimation in more detail.

--As one the reviewers has mentioned please define the method of denominator estimation in more detail.

--Reviewer #4: I think it is necessary to define the method of denominator estimation in more detail. This is especially true considering emigration or death which ordinarily happens in populations.

Thank you for your comment. In the methods section, we originally stated that the relevant analysis was based solely on individuals who could be tracked for 5 years, which is a simple and straightforward approach without additional conditions. In the discussion, we noted that individuals who left the insurance system, such as those who moved, were excluded by this method. However, as per your suggestion, we have now added a mention of those exclusions in the methods section for clarity.

“Subsequent analyses included insured individuals who could be continuously tracked for 5 years from FY2015 to FY2019, excluding those who left the insurance system for reasons such as moving or had gaps in their tracking data, while still including those who died during this period.”

--Reviewer #4: Please define the method of confidence interval estimation.

We used R and used a standard generalized linear model with weights and computed confidence intervals based on the standard errors of the coefficients. We made this revision.

“Finally, a generalized linear model weighted by the denominator was constructed to analyze the annual increase in prevalence, considering age, sex, and their interaction (R 4.4.1). Confidence intervals for the coefficient was estimated using standard errors derived from the weighted model.”

There was one instance in the previous revision where a terminology update was missed, which may have caused some confusion. We have now ensured consistency in the terms related to the confidence interval.

“The model, accounting for age and sex, also estimated the annual incidence at 3.03% (95% CI: 2.21%–3.85%).”

---

## [Editor Report · Decision Letter 2]

20 Sep 2024

Estimating diabetes mellitus incidence using health insurance claims data: A database-driven cohort study

PONE-D-24-14955R2

Dear Dr. Kunisawa,

We’re pleased to inform you that your manuscript has been judged scientifically suitable for publication and will be formally accepted for publication once it meets all outstanding technical requirements.

Kind regards,

Hamid Reza Baradaran, M.D., Ph.D.,

Academic Editor

PLOS ONE
---

## [Editor Report · Acceptance letter]

24 Sep 2024

PONE-D-24-14955R2 

PLOS ONE

Dear Dr. Kunisawa, 

I'm pleased to inform you that your manuscript has been deemed suitable for publication in PLOS ONE. Congratulations! Your manuscript is now being handed over to our production team.

Kind regards, 

on behalf of

Professor Hamid Reza Baradaran 

Academic Editor

PLOS ONE